# Effect of Dietary Forage: Concentrate Ratio on Pre-Caecal and Total Digestive Tract Digestibility of Diverse Feedstuffs in Donkeys as Measured by the Mobile Nylon Bag Technique

**DOI:** 10.3390/ani10061070

**Published:** 2020-06-20

**Authors:** Li-Lin Liu, Xiao-Ling Zhou, Hong-Jian Yang

**Affiliations:** 1State Key Laboratory of Animal Nutrition, College of Animal Science and Technology, China Agricultural University (CAU), Beijing 100193, China; llltaru@163.com; 2College of Animal Science, Key Laboratory of Tarim, Animal Husbandry Science and Technology, Xinjiang Production and Construction Group, Tarim University, Alar 843300, China; zxldky@126.com

**Keywords:** donkey, pre-caecal digestibility, total digestive tract digestibility, mobile nylon bag

## Abstract

**Simple Summary:**

More and more researchers are paying attention to the nutrition of donkeys, but the research on the digestion of feed in the prececum and the whole digestive tract of donkeys is very rare. In this study, a Latin square experimental design was applied to determine the effect of dietary forage: concentrate (F:C) ratio on pre-caecal and total digestive tract digestibility of four feedstuffs in donkeys was measured by the mobile nylon bag technique. High-forage diets resulted in the lowest mean retention time of bags at the ileo-caecal junction. Starchy corn meal in comparison with soybean meal encountered greater extent of prececum digestion and prececum fiber digestion in fibrous forages contributed over 50% percent of total tract digestion. The nutrient composition of the feed, especially the fiber content was the main factor that affects the digestibility contribution of the prececum.

**Abstract:**

The domestic donkey is a unique equid species with specific nutritional requirements, however, limited laboratory evidences are available to address the digestibility contribution of the prececum in relation to the total digestive tract. In the present study, six caecum-fistulated adult female Xinjiang donkeys served as the experimental animals in a 3 × 3 Latin square design, and mobile nylon bag technique was applied to determine the effect of dietary F:C ratio on pre-caececum and total digestive tract digestibility of rice straw, alfalfa hay, corn meal, and soybean meal. The dietary treatments included: (1) HF, a high-fiber ration (F:C = 80:20), (2) MF, a medium-fiber ration (F:C = 55:45), and (3), LF, a low-fiber ration (F:C = 35:65). The experiment consisted of three consecutive Latin square periods, and each period lasted 25 days. In each period, the animals were administrated naso-gastrically nylon bags (38 μm pore size) containing aforementioned feeds. After 1.5 h intubation, the bags were checked once an hour and collected at the ileo-caecal junction (small intestine bag, D1) and in the feces (fecal bag, D2). Regardless whatever feeds were introduced, the percentage of bag collected (BC) was quadratically (HF) or linearly (MF and LF) increased against different fixed bag collection time. The highest BC occurred in MF (73.8%), but no significant difference was observed between HF (62.3%) and LF (50.8%). The lowest mean bag retention time was observed in HF (2.7 h), and no significant difference occurred between MF (4.6 h) and LF (5.0 h) diets. For each feed, D1 and D2 digestibility for DM, CP, NDF, and ADF did not differ among three dietary treatments (*p* > 0.05). Regardless of whatever diets were fed to the donkeys, D2 digestibility for DM and CP among the feeds ranked as: soybean meal > corn meal > alfalfa hay > rice straw (*p* < 0.01). D1 digestibility for DM among the feeds ranked as: corn meal > soybean meal > alfalfa hay > rice straw (*p* < 0.01). D1 digestibility for CP among the feeds ranked as: soybean meal > corn meal > alfalfa hay > rice straw (*p* < 0.01). In summary, dietary forage: concentrate ratio did not affect pre-caecal or total tract nutrient digestibility. The fiber level in feeds was the main limiting factor to affect the digestibility contribution of the pre-caecum in relation to the total digestive tract.

## 1. Introduction

Horses and donkeys, belonging to *Equidae* family, can digest a high proportion of dietary starch, proteins, and fats via enzymatic activity in the small intestine though they are well recognized as the hindgut fiber fermentation type livestock [1,2,3]. However, diets containing high levels of cereals can pre-dispose horses to diet-related metabolic disorders such as acidosis, colic, and laminitis [4,5]. Horses fed with a low-fiber concentrated diet have higher prevalence and higher severity of equine gastric ulceration syndrome in comparison with horses on pasture [6]. Conversely, horses consuming fiber-based diets are less susceptible to acidosis because dietary fiber can maintain more stable hindgut fermentation variables than cereals [7,8].

Xinjiang donkey is a small breed that is mainly distributed around Kashi, Hotan, and other areas of southern Xinjiang province in China [9]. For a long time, most donkeys fed on fibrous feeds are raised for pack transport, pulling carts, farm tillage, raising water, or milling in the world. In recent years, the donkey raising industry has developed rapidly, catering for their use in meat and milk production. National Research Council (2007) provides dry matter intake, ration formulation, and nutrient allowance recommendations (e.g., digestible energy, protein, calcium, and phosphorus) for donkeys fed on good or poor forage quality, which assume certain digestibility coefficients for energy and protein.

The small intestine is generally considered the main site of enzymatic hydrolysis of starch and protein while dietary fiber is mainly subjected to microbial fermentation in hindguts of caecum and large intestine. However, lack of information on the site of digestion in the prececum and total digestive tract of donkeys sheds some doubt on the accuracy of these formulations. The digestibility of a given feedstuff is influenced by botanic variety of forage, energy, and protein supplement feeds. In the past decades, the mobile nylon bag technique (MNBT), originally applied in ruminant animals [10], has been successfully used in cecum fistulated horses to determine pre-cecal digestibility of various feeds [11,12,13,14,15,16,17]. In contrast, no literature with MNBT in donkeys is available so far to address pre-caecal digestibility of various feeds, and it is also not clear if dietary forage: concentrate (F:C) ratio could alter the digestive role of different site in donkey gastrointestinal tract. In the present study, MNBT was first introduced in Xinjiang donkeys, and the objective was to determine effect of dietary F:C ratio on pre-caecal and total digestive tract digestibility of diverse feedstuffs in donkeys.

## 2. Materials and Methods

In the present study, all of the procedures performed in animal feeding and sample collection followed the Guidelines of the Beijing Municipal Council on Animal Care (with protocol AW23050202-1).

### 2.1. Animals

Six adult female Xinjiang donkeys weighed 180 ± 10 kg were served as experimental animals, and each animal free access to water was housed in a separate pen (2 m × 5 m). Cecal fistula (Figure 1: Inner diameter 40 mm, length 50 mm) were surgically installed near the ileocecal junction after three-day fasting. During the body health recovery period, the animals free access to water and vitamin-mineral blocks were ad libitum fed alfalfa hay and rice straw at a ratio of 3:1 until feed intake returned to normal amount. In the present study, it took approximate 3 months for these animals to return to normal health status.

### 2.2. Experimental Design and Bag Collection Procedure

A replicate 3 × 3 Latin square experimental design was applied for six cecum-fistulated donkeys to determine the effect of three dietary treatments differing in forage:concentrate (F:C) ratio on pre-caecal and total digestive tract digestibility of rice straw, alfalfa hay (mid-bloom), corn meal, and soybean meal (Table 1) with mobile nylon bag technique. As shown in Table 2, the dietary treatments included: (1) HF, a high-fiber ration (F:C = 80:20), (2) MF, a medium-fiber ration (F:C = 55:45), and (3), LF, a low-fiber ration (LF: F:C = 35:65). The experiment consisted of three Latin square periods, and each period lasted 25 days including 18 days for diet adaption, 3 days for the introduction of mobile nylon bags and 4 days for the bag collection from cecum fistula or feces. In each period, two of six animals were randomly allocated to one of the three dietary treatments, and each animal was housed in a separate pen and the animals had free access to water. The forage was chopped into approximately 1 cm lengths and then wetted by adding 30% water. Afterwards the forage was mixed with the concentrate to prepare total mixed rations. According to feed intake recovery prior to the start of the experiment, all animals were provided 1.2 times amount of each corresponding ration (dry matter intake = 4.0±0.3 kg), and divided into three portions and fed ad libitum at 08:30, 13:30, and 19:00, respectively.

On the day 19 of each period, mobile nylon bags (Figure 2: 1 cm diameter, 6 cm length, 38 μm pore size) containing weighed feeds (2.0 mm sieve size, 500 mg of the tested forage or 700 mg of the tested corn meal or soybean meal) and a steel washer (9 mm diameter, 3 mm thickness) were individually introduced into the stomach via esophagus of each animal with a 150 cm length nasogastric tube; a magnetic bar (double-sided NdFeB, 60 mm × 20 mm) was hanged inside the cecum fistula. After 1.5 h of the bag introduction [16], the lips of fistulas were opened once an hour until 10.5 h, and the bags arrived near the ileocecal junction were captured with the magnetic bar and recorded to determine pre-cecum digestibility (D1). After the pre-caecal bag collection, the magnet bar was removed away from inside of the cecum fistula, and the remaining bags were collected in feces to determine total tract digestibility (D2). In the above procedure, 10 bags per feed per animal were introduced for each determination, and the recovered bags were immediately stored at −20 °C for later laboratory washing and chemical analysis.

### 2.3. Chemical Analysis

All the recovered bags were thawed at room temperature and washed with tap water until the water became clear. Afterwards, they were dried at 65 °C for 48 h and weighed at room temperature. The feed residues left inside bags were collected and pooled for same feed within an animal and subjected to chemical analysis. Representative samples of the tested feeds and their residues collected from mobile nylon bags were analyzed. The samples were dried at 105 °C for 4 h to determine the dry matter (DM). Crude protein (CP) was determined by Kjeldahl method (N × 6.25) following the Association of Official Analytical Chemists (AOAC Official Method 2001.11) [18]. Neutral detergent fiber (NDF) and acid detergent Fiber (ADF) were determined using an automatic fiber analyzer (A2000i, Ankom Technology, Macedon, NY, USA) following the method as described by Van Soest PJ et al. (1991) [19]. In the NDF analysis, heat-stable α-amylase was applied. Both NDF and ADF contents were corrected with the residual ash content.

### 2.4. Calculations and Statistical Analysis

The percentage of bags collected (BC) in each animal was calculated for each tested feed during each Latin square period. In addition, the transit characteristics of mobile nylon bags for each dietary treatments were determined by calculating the cumulative percentage of bags collected at each fixed collecting time. The mean retention time (MRT) of bags in pre-caecal tract was calculated according to Faichney (1975) [20]:MRT = (ΣB_i_ × Δt_i_)/ΣB_i_(1)
where B_i_ is the number of bags collected at time t_i_, Δt_i_ is average time since bag administration and calculated as follows:t_i_ = (t_i_ − t_0_) + (t_i_ + t_0_)/2(2)
where t_0_ is the initial time of bag administration, and ti is the time when bags were captured near the ileocecal junction.

Pre-cecum digestibility (D1) and total digestive tract digestibility (D2) were calculated according to nutrient difference between initially introduced nutrient mass and residual nutrient mass recovered from the bags collected from cecum fistula and feces, respectively.

The BC data against different collection time in each dietary treatments was subjected to linear or quadratic regression analysis with the REG procedure of SAS software (version 9.2; SAS institute Inc., Cary, NC, USA). The results of regression equations, correlation coefficient (r^2^) and statistical *p* value are presented in Figure 3.

Except the above regression analysis, data on prececum and total tract digestibility were analyzed using the mixed procedure of the SAS software. The statistical model was applied as follows:*Y_ijk_* = μ + *R_i_* + *F_j_* + (*R* × *F*)*_ij_* + *P_k_* + *A_l_* + *e**_ijkl_*(3)
where *Y_ijk_* is the dependent variable, μ is the overall mean, *Ri* is the ration effect (HF, MF, LF), *Fj* is the feed effect (rice straw, alfalfa hay, corn meal, soybean meal), *R* × *F* is the interaction effect between ration and feed, *P_k_* is the period effect (*k* = 3), A*_l_* is the animal random effect (*l* = 6), and *e_ijkl_* is the error term. First-order autoregressive and compound symmetry (homogeneous and heterogeneous) were tested as covariance structures, and the covariance structure with the least Akaike information criterion was retained in the final model. Sums of squares for treatment were separated into single-degree-of-freedom preplanned orthogonal contrasts. Least squares means and standard errors (SEM) were reported and compared between rations or between feeds with a multiple comparison test (Tukey/Kramer). Significance was declared at *p* < 0.05. Probability values between 0.05 and 0.10 were considered as trends.

## 3. Results

### 3.1. Bag Collection Rate and Mean Retention Time

Regardless of the kind of feed that was introduced, the percentage of bag collected at cecum was step-wise increased against the collecting time (Figure 3). In both MF and LF group, BC increased against the time. BC in HF group increased rapidly in the first 3.5 h, and its increase became slow in the subsequent period. As a result, the highest average BC across the whole collecting time occurred in MF, and the lowest occurred in LF (*p* = 0.04, Table 3). The lowest MRT was observed in HF, no significant difference occurred between MF and LF group.

### 3.2. Pre-Cecum and Total Digestive Tract Digestibilities of Feeds

Regardless of whatever diets were fed to the donkeys, as shown in Table 4, D1, D2, and D1/D2 values for DM, CP, NDF, and ADF varied among the feeds. Because of the high DM digestibility of corn meal and soybean meal, no enough residues were collected for the determination of NDF and ADF and resulted in the missing of D1 and D2 for these feeds.

Regardless of whatever feeds were tested, D1 and D2 values for DM, CP, NDF, and ADF did not differ among three dietary treatments. Consequently, no significant difference between dietary treatments occurred for the digestion contribution of pre-caecal tract in total digestive tract (D1/D2). As for the rice straw, mean D1/D2 ratios were 0.79, 0.71, 0.62, and 0.65 for DM, CP, NDF, and ADF respectively. As for alfalfa hay, mean D1/D2 ratios were 0.68, 0.85, 0.51, and 0.57 for DM, CP, NDF, and ADF respectively. As for corn meal, mean D1/D2 ratios were 0.94 and 0.88 for DM and CP, respectively. As for soybean meal, mean D1/D2 ratios were 0.86 and 0.98 for DM and CP, respectively.

## 4. Discussion

### 4.1. Effect of Diet Type on Bag Collection Rate

Currently, there are two efficient methods reported mainly for horses how to introduce mobile nylon bag into stomach in a short time. First, nylon bag is delivered to the stomach of the horse mainly by flushing the nylon bag with water into the stomach through the nasogastric tube, Second, nylon bag along the nasogastric tube was blow into the stomach using a small air pump [14]. However, the esophagus of donkeys is relatively narrow, which limit the use the nasogastric tube with 1.8 cm external diameter as commonly applied in horses [15]. To overcome such limitation, 1 cm external diameter nasogastric tube was used in the present study, and nylon bags were one by one pushed into the stomach with a guide wire. All donkeys in the present study remained healthy throughout the experiment, and intubations of nylon bags went smoothly without any indication of choking or constipation. As a result, the average BC value in the present study during 10.5 h collecting time ranged 50.8–73.8%, and it was very close to the results (50–75%) with ponies [12]. The BC value in the present study was greater than 37% in horses over a 7 h collection period [16], but less than the value of 73% reported in other equid animals with a 13 h collection period [17] and 79% reported in horses with a 13 h collection period [21]. Our BC results suggest that the recovery rate of bags will increase when donkeys were fed high-fiber rations, and this could be due to high flow rate of the bags in the gastrointestinal tract, pushed by the high-fiber fraction in chime movement.

### 4.2. Effect of Diet Type on Mean Retention Time

Donkey have a well-developed digestive tract, yet unlike cattle, sheep, and goats, donkeys do not possess rumen. Once their food is ingested it will eventually go through a functional caecum. The functional caecum is at the posterior end of the digestive system and is responsible for additional break down of nutrients through bacterial fermentation. In the present study, the MRT of nylon bags pre-cecal was observed 4.1 h, which was close to 4.2 h ± 0.6 observed in horses [22], greater than 3.3 h observed in ponies [14] and 3.6 h observed in horses [16], and less than <6.9 h observed in horses [17]. So far, no pre-cecal retention time has been reported previously for donkeys. In the present study, the lowest MRT in Xinjiang donkeys occurred in HF (2.7 h), and no difference was observed between MF (4.6 h) and LF (5.0 h), implicating that comparative low-fiber chyme could not facilitate the movement of the bags in the gastrointestinal tract, resulting in the increase of bag retention time in pre-cecum tract.

### 4.3. Effect of Diet Type on Feed Digestibility in The Prececum and Total Digestive Tract

The mobile bag technique used on caecum-fistulated donkeys offers the opportunity to measure the degree of individual feedstuff degradation within different compartment of the alimentary tract. The knowledge of digestibility contribution of a certain segment of the gut is particularly important for different feeds when considering energy production. Although dietary protein can be digested via enzymatic process in small intestine, it is also very useful to know how much undigested protein was further fermented in the large intestine to ammonia N, which is an essential factor for the growth and maintenance of the hindgut microbial population if dietary fiber component is to be well utilized [5,14]. As expected, the starch-rich corn meal and protein-rich soybean meal presented higher pre-caecal digestibility (D1) as well as total tract digestibility (D2) in terms of DM and CP than alfalfa hay and rice straw though both D1 and D2 were not affected by dietary forage: concentrate ratio. The differences between D2 and D1 of DM accounted to a few percent units only for corn meal (3.8–8.7%) in comparison with soybean meal (12–8.7%), suggesting that the starch-rich energy feed in comparison with the protein supplemental feedstuff encounter greater prececum enzymatic digestion and less microbial hindgut fermentation. The results were in agreement with Meyer et al. (1995) [23] and Julliand et al. (2006) [24] who conducted starch digestibility in the prececum and total digestive tract of horses.

Plant fiber fermentation could theoretically happen throughout the equine gastrointestinal tract though the soluble fiber was more digestible than the insoluble fiber fraction [17]. In the present study, although both D1 and D2 in terms of DM, CP, NDF, and ADF were greater in alfalfa hay and rice straw as expected, interestingly, the differences between D2 and D1 of DM accounted to high percent units for alfalfa hay (13.7–20.4%) in comparison with rice straw (5.1–7.3%). Similar high percent unit differences also occurred for CP, NDF, and ADF digestibility in alfalfa hay compared with rice straw, and these results implicated that prececum digestion also contribute over 50% percent of total tract digestion of fiber originating from fibrous feedstuffs, and the degree of total tract digestibility is feed-specific dependent though lack of fibrolytic enzyme activity was commonly believed to exist in the prececum digestion. This could explain why horses or donkeys can fulfill their energy demands for physical activity when only fed high-quality forage. In the case of donkey’s meat production instead of the work use, animal producers or nutritionist should keep in mind that it is important to partially substitute poor-quality forage with good-quality forage, cereal grain, and protein supplement.

## 5. Conclusions

Independent of whatever diets fed to the donkeys, the starch-rich corn meal in comparison with soybean meal encounters greater prececum enzymatic digestion and less microbial hindgut fermentation. Regarding the fibrous feedstuffs, prececum fiber digestion contributes to over 50% of the total tract digestion, and the fiber content of feeds was the main factor affecting the digestibility contribution of the prececum in relation to the total digestive tract, though the lack of fibrolytic enzyme activity in the prececum digestion is commonly believed to be the cause.

## Figures and Tables

**Figure 1 animals-10-01070-f001:**
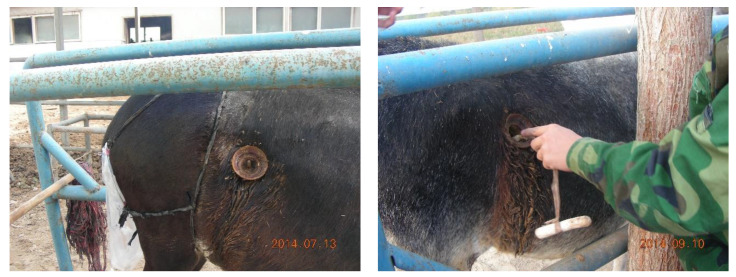
Cecal fistula (inner diameter 40 mm, length 50 mm) were surgically installed near the ileocecal junction, and a magnetic bar was put into cecal fistula and hanged near the ileocecal junction of each donkey.

**Figure 2 animals-10-01070-f002:**
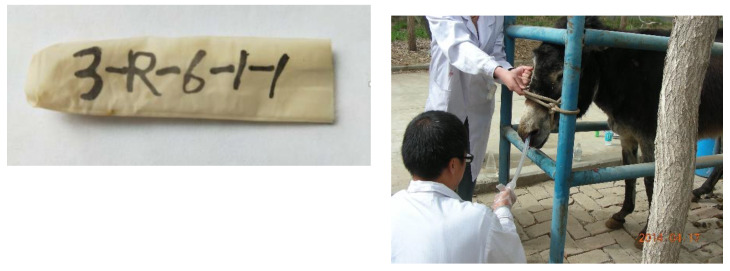
Mobile nylon bags (1 cm diameter, 6 cm length, 38 μm pore size) and a steel washer (9 mm diameter, 3 mm thickness) containing weighed feeds (2.0 mm sieve size) and a steel washer (9 mm diameter, 3 mm thickness) were individually introduced into the stomach via esophagus of each animal with a 150 cm length nasogastric tube.

**Figure 3 animals-10-01070-f003:**
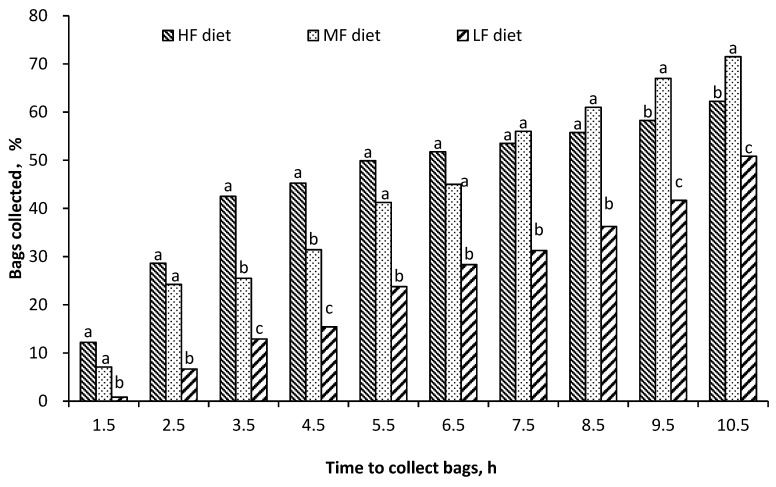
Cumulative collection percentage of mobile nylon bags from the cecum of donkeys fed high-fiber (HF), medium-fiber (MF), or low-fiber (LF) rations. The regression analysis of bag collection percentage (y) against the time of bag collected showed as follows: HF, y = −0.70 × ^2^ − 12.99 × −0.69, r^2^ = 0.26, *p* < 0.001; MF, y = 7.73 × −6.91, r^2^ = 0.75, *p* < 0.001; LF, y=5.23 × −6.62, r^2^ = 0.73, *p* < 0.001 within a time of collection; bars without a common superscript letter differ at *p* < 0.05.

**Table 1 animals-10-01070-t001:** Chemical composition in dry matter of target feedstuffs applied for cecal disappearance test.

Item	Rice Straw	Alfalfa Hay	Corn Meal	Soybean Meal
Crude protein (g/kg)	51	121	88	470
Neutral detergent fiber (g/kg)	787	693	117	135
Acid detergent fiber (g/kg)	643	543	36	71

**Table 2 animals-10-01070-t002:** Feed ingredients and nutrient composition of high-(HF), medium-(MF) and low-fiber (LF) rations fed to experimental donkeys.

Item	HF	MF	LF
Feed ingredient			
rice straw (g/kg)	700	450	250
alfalfa hay (g/kg)	100	100	100
corn meal (g/kg)	0	150	352
soybean meal (g/kg)	33	50	54
cottonseed meal (g/kg)	150	225	217
premix feed (g/kg)	17	25	27
Forage:Concentrate	80:20	55:45	35:65
Nutrient level in dry matter basis			
DE (MJ/kg)	10.07	11.91	13.85
Crude protein (g/kg)	122	168	178
Neutral detergent fiber (g/kg)	593	451	328
Acid detergent fiber (g/kg)	469	346	238
NFC (g/kg)	116	235	371
NDF/NFC	5.1	1.9	0.9

**Table 3 animals-10-01070-t003:** Average percentage of bags collected (BC) and mean retention time (MRT) of mobile nylon bags recovered at ileocecal junction site of donkeys fed high- (HF), medium- (MF), and low-fiber (LF) rations.

Items	HF	MF	LF	S.E.M	*p* Value
BC (%)	62.3 ^a,b^	73.8 ^a^	50.8 ^b^	4.90	0.04
MRT (h)	2.7 ^b^	4.6 ^a^	5.0 ^a^	0.55	<0.01

^a,b^ Values in a row without same lowercase superscript letter differ between diets at *p* <0.05; SEM, standard error of the mean.

**Table 4 animals-10-01070-t004:** Pre-cecum (D1) and total digestive tract (D2) digestibility coefficients of different feeds for Xinjiang donkeys fed high-(HF), medium-(MF), and low-fiber (LF) rations.

ITEMS	Feeds	Ration				S.E.M	*p* Value		
		HF	MF	LF	Average ^1^		Ration	Feed	R × F
D1 of DM, %	RS	24.0	23.6	24.6	24.0 ^D^	1.53	0.05	<0.01	0.16
	AH	36.2	36.5	37.0	36.5 ^C^				
	CM	83.0	88.2	89.0	86.7 ^A^				
	SBM	80.2	85.5	80.5	82.1 ^B^				
	average ^2^	57.6	60.4	59.6					
D2 of DM, %	RS	30.8	28.7	31.9	30.4 ^C^	2.31	0.30	<0.01	0.09
	AH	42.9	56.9	50.7	51.1 ^B^				
	CM	91.7	93.3	92.8	62.6 ^A^				
	SBM	92.5	97.5	94.7	94.9 ^A^				
	average ^2^	66.4	69.1	67.5					
D1/D2for DM	RS	78.4	82.5	77.2	79.3 ^C^	3.02	0.95	<0.01	0.14
	AH	84.6	73.3	82.2	79.5 ^C^				
	CM	90.4	94.5	95.8	93.6 ^A^				
	SBM	86.7	87.7	85.1	86.5 ^B^				
	average ^2^	85.1	84.5	85.1					
D1 of CP, %	RS	56.1	53.3	50.6	53.4 ^D^	3.31	0.92	<0.01	0.58
	AH	83.0	77.4	72.5	79.6 ^C^				
	CM	83.4	88.3	89.2	87.0 ^B^				
	SBM	95.6	97.0	96.0	96.3 ^A^				
	average ^2^	79.5	79.0	78.6					
D2 of CP, %	RS	44.3	33.9	32.6	35.6 ^D^	1.86	0.35	<0.01	<0.01
	AH	76.7	74.2	69.3	72.3 ^C^				
	CM	91.1	90.8	91.4	91.1 ^B^				
	SBM	97.4	99.5	98.3	98.4 ^A^				
	average ^2^	80.2	79.8	78.2					
D1/D2 for CP	RS	127.0	157.4	155.5	146.6 ^A^	5.02	0.15	<0.01	0.11
	AH	108.6	104.6	104.8	105.9 ^B^				
	CM	91.7	97.4	97.7	95.6 ^C^				
	SBM	98.2	97.6	97.7	97.8 ^C^				
	average ^2^	106.3	114.3	113.9					
D1 of NDF, %	RS	16.7	16.6	17.0	16.8 ^B^	2.08	0.79	<0.01	0.86
	AH	22.8	23.7	25.2	23.9 ^A^				
	average ^2^	19.7	20.1	21.1					
D2 of NDF, %	RS	27.1	24.6	29.3	27.0 ^B^	3.67	0.12	<0.01	0.16
	AH	31.6	44.7	47.2	42.0 ^A^				
	Average ^2^	28.9	32.6	38.2					
D1/D2for NDF	RS	61.9	68.5	57.9	62.7	5.25	0.40	0.23	0.22
	AH	65.8	50.9	54.1	56.5				
	average ^2^	63.4	61.4	56.0					
D1 of ADF, %	RS	18.3	12.7	14.0	14.5 ^B^	4.31	0.68	<0.01	0.80
	AH	28.3	26.0	30.3	28.1 ^A^				
	average ^2^	23.3	19.3	22.1					
D2 of ADF,%	RS	24.9	20.3	23.8	22.9 ^B^	3.77	0.05	<0.01	0.02
	AH	31.2	50.7	52.1	46.3 ^A^				
	average ^2^	27.4	35.3	37.9					
D1/D2 for ADF	RS	67.9	63.8	58.4	62.7	4.29	0.34	0.08	0.35
	AH	58.7	50.3	57.8	54.7				
	average ^2^	64.8	57.0	58.1					

^1 A–D^ Values in a column without same uppercase superscript letter differ between feeds at *p* <0.05. ^2^ Values in a row without same lowercase superscript letter differ between diets at *p* <0.05. RS, rice straw; AH, alfalfa hay; CM, corn meal; SBM, soybean meal; DM, dry matter; CP, crude protein; NDF, neutral detergent fiber; ADF, acid detergent fiber; SEM, standard error of the mean.

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
