# Peer review of "Effect of Dietary Forage: Concentrate Ratio on Pre-Caecal and Total Digestive Tract Digestibility of Diverse Feedstuffs in Donkeys as Measured by the Mobile Nylon Bag Technique"

_animals, 2020, doi:10.3390/ani10061070_

Round 1

Reviewer 1 Report

Please answer my questions and comments.
L50; the effect of dietary F:C ratio on nutrient digestion in the foregut were not significant though retention time and passage rate was modified. --- Donkeys belonging to hindgut fermentation type did not affect nutrient digestion in the foregut irrespective of dietary F:C ratio. What is new ?
L51; The forage fibre content was main limiting factor, which affect the digestibility contribution of the foregut in relate to the total digestive tract. --- I can expect easily from existing knowledge. What is new ?
L78-; In the present study,... --- The problem to be solved is not clear before this.
L181; Our BC results suggest that the recovery rate of bags will increase when donkeys were fed high-fibre rations. --- Can you tell me what is the reason ?
L191; chime production --- I cannot understand this. Is this the reason why decreasing dietary fibre content will lead to increase of the bag retention time of feeds in pre-cecum tract ? Can you tell me why ?
L218; V. Julliand a and A. De Fombelle b (2006) [26] --- Julliand, et al. (2006) [26]
L219; of horses . --- delete space
L229; And this could explain why horses or donkeys can fufil their energy demands for physical activity when only fed high-quality grass hay or even rough forage like rice straw. --- You notice only differences between D2 and D1. I think absolute values of DM digestibility (D2) in rice straw 30.8, 28.7 and 31.9% are very low compared to corn meal 91.7, 93.3 and 92.8% (and also soybean meal). Can you say so ?
L229; donkeys can fufil their energy --- fulfill ?
L232; it is import to replace --- it is important to replace ?
L232; with particular aim to satisfy their use in pharmaceutical, meat, and milk production. --- I cannot understand this meaning.
Conclusions
Can you explain the reason why foregut fibre digestion contributes over 50% of total tract digestion though lack of fibrolytic enzyme activity was commonly believed to exist in the foregut digestion ?
L240; botanical origin --- What do you mean ? Are corn meal and soybean meal not botanical origin ?
L251; Canadian Veterinary Journal La Revue Veterinaire Canadienne --- Can. Vet. J. La Revue Veterinaire Canadienne
L258; Acta Veterinaria Brno --- Acta Vet. Brno
L260; In Proceedings of the 12th Equine Nutrition and Physiology Symposium --- In Proc. the 12th Equine Nutr. Physiol. Symp.
L262; Animal Feed Science and Technology --- Anim. Feed Sci. Tech.
L264; Animal Science --- Anim. Sci.
L270; British Society of Animal Science --- Proceedings of the British Society of Animal Science, BSAS, Penicuik, Midlothian, UK. ? Proc. Br. Soc. Anim. Sci., BSAS, Penicuik, Midlothian, UK. ?
L278; Animal Feed Science and Technology --- Anim. Feed Sci. Tech.
L280; Arch Anim Nutr --- Arch. Anim. Nutr.
L285; Animal Science --- Anim. Sci.
L287; J Anim Physiol Nutr (Berl) --- J. Anim. Physiol. Nutr. (Berl.)
L296; Livestock Science --- Livest. Sci.
L298; Acoustics Speech & Signal Processing Newsletter --- Acoust. Speech Signal Process. Newslett. IEEE
L301; Livestock Science --- Livest. Sci.
V. Julliand a, *.; A. De Fombelle b, M.b. --- Julliand V.; De Fombelle A.; Varloud M.

Author Response

Please answer my questions and comments.

Point 1:  L50; the effect of dietary F:C ratio on nutrient digestion in the foregut were not significant though retention time and passage rate was modified. --- Donkeys belonging to hindgut fermentation type did not affect nutrient digestion in the foregut irrespective of dietary F:C ratio. What is new ?

Response 1: The sentence has been revised as you suggested. The current study reported the use of mobile bag technique in donkeys for the first time.

Point 2:  L51; The forage fibre content was main limiting factor, which affect the digestibility contribution of the foregut in relate to the total digestive tract. --- I can expect easily from existing knowledge. What is new ?

Response 2: Yes, it is easily understood based on existing knowledge, the present study confirmed that knowledge for the first time based on mobile bag technique in donkeys, implicating that the small intestine is indeed considered the main site of enzymatic hydrolysis of nutritients in donkeys like in horses.

Point 3:  L78-; In the present study,... --- The problem to be solved is not clear before this.

Response 3:  The whole paragraph has been revised now, and hopefully the objective is reasonably described.

 Point 4:  L181; Our BC results suggest that the recovery rate of bags will increase when donkeys were fed high-fibre rations. --- Can you tell me what is the reason ?

Response 4: This could be due to high flow rate of the bags in the gastrointestinal tract, pushed by the high-fibre fraction in chime movement.

 Point 5:  L191; chime production --- I cannot understand this. Is this the reason why decreasing dietary fibre content will lead to increase of the bag retention time of feeds in pre-cecum tract ? Can you tell me why ?

Response 5: As noted in the above response, this could be due to high flow rate of the bags in the gastrointestinal tract, pushed by the high-fibre fraction in chime movement. The low-fibre chyme could not facilitate the movement of the bags in the gastrointestinal tract. 

 Point 6:  L218; V. Julliand a and A. De Fombelle b (2006) [26] --- Julliand, et al. (2006) [26]

*Response 6: Yes, accepted and revised.

Point 7:  L219; of horses . --- delete space

Response 7: Yes, accepted and deleted now.

 Point 8:  L229; And this could explain why horses or donkeys can fufil their energy demands for physical activity when only fed high-quality grass hay or even rough forage like rice straw. --- You notice only differences between D2 and D1. I think absolute values of DM digestibility (D2) in rice straw 30.8, 28.7 and 31.9% are very low compared to corn meal 91.7, 93.3 and 92.8% (and also soybean meal). Can you say so ?

Response 8: Yes, that should be true only in the case of high-quality forage instead of poor-quality forage like rice straw. The sentence has been revised now by deleting “or even rough forage like rice straw”

 Point 9:  L229; donkeys can fufil their energy --- fulfill ?

Response 9: Yes, accepted and deleted now.

 Point 10:  L232; it is import to replace --- it is important to replace ?

Response 10: Yes, accepted and deleted now.

 Point 11:  L232; with particular aim to satisfy their use in pharmaceutical, meat, and milk production. --- I cannot understand this meaning.

Response 11: It is important to partially substitute poor-quality forage with good-quality forage, cereal grain and protein supplement when donkey raising shifts from the work use to the use to produce meat and milk. The sentence has been revised now

Conclusions

Point 12: Can you explain the reason why foregut fibre digestion contributes over 50% of total tract digestion though lack of fibrolytic enzyme activity was commonly believed to exist in the foregut digestion ?

Response 12: The results obtained in the present study implicate that partially microbial fermentation might also occur in the foregut though lack of fibrolytic enzyme activity was commonly believed to exist in the foregut digestion.   

 Point 13: L240; botanical origin --- What do you mean ? Are corn meal and soybean meal not botanical origin ?

Response 13: It has been revised to ‘botanical nature’ which reflect different plant variety.

 Point 14:  L251; Canadian Veterinary Journal La Revue Veterinaire Canadienne --- Can. Vet. J. La Revue Veterinaire Canadienne

Response 14: Yes, revised now.

 Point 15:  L258; Acta Veterinaria Brno --- Acta Vet. Brno

Response 15: Yes, revised now.

Point 16:  L260; In Proceedings of the 12th Equine Nutrition and Physiology Symposium --- In Proc. the 12th Equine Nutr. Physiol. Symp.

Response 16: Yes, revised now.

Point 17:  L262; Animal Feed Science and Technology --- Anim. Feed Sci. Tech.

Response 17: Yes, revised now.

Point 18:  L264; Animal Science --- Anim. Sci.

Response 18: Yes, revised now.

Point 19:  L270; British Society of Animal Science --- Proceedings of the British Society of Animal Science, BSAS, Penicuik, Midlothian, UK. ? Proc. Br. Soc. Anim. Sci., BSAS, Penicuik, Midlothian, UK. ?

Response 19: Yes, revised now.

 Point 20:  L278; Animal Feed Science and Technology --- Anim. Feed Sci. Tech.

Response 20: Yes, revised now.

 Point 21:  L280; Arch Anim Nutr --- Arch. Anim. Nutr.

Response 21: Yes, revised now.

 Point 22:  L285; Animal Science --- Anim. Sci.

Response 22: Yes, revised now.

Point 23:  L287; J Anim Physiol Nutr (Berl) --- J. Anim. Physiol. Nutr. (Berl.)

Response 23: Yes, revised now.

 Point 24:  L296; Livestock Science --- Livest. Sci.

Response 24: Yes, revised now.

Point 25:  L298; Acoustics Speech & Signal Processing Newsletter --- Acoust. Speech Signal Process. Newslett. IEEE

Response 25: Yes, revised now.

Point 26:  L301; Livestock Science --- Livest. Sci.

  1. Julliand a, *.; A. De Fombelle b, M.b. --- Julliand V.; De Fombelle A.; Varloud M.

Response 26: Yes, revised now.

Reviewer 2 Report

This paper deals with the digestibility evaluation of 4 feedstuffs under three diet managements using the mobile bag technique in donkeys. The mobile bag technique has been used during decades for digestibility assays with different animal species and digestibility values for the studies feedstuffs has been analysed elsewhere. Therefore, the novelty of this paper is not very high apart from the animal species used.

Overall the paper is well written, but English should be revised. Only significant results should be address and not discuss numerical changes. The statistical model should be improved. Discussion must be improved.

Introduction

In the introduction there is a lack of background arguments to hypothesize that the forage to concentrate ratio could affect retention time or digestibility of tested feeds.

You should include a cleared objective for your study. You commented in the intro that the problem is that the rations actually are formulated following NRC, which assumes some digestibility values and in donkeys would be different. Therefore, I don´t understand the objective of the study because digestibility references were obtained under standard feeding regimes, and there is no justification for the inclusion of different forage to concentrate ratios.

Materials and methods

The four feedstuffs included in the study were usually used in donkey’s rations? They are the most usually used feedstuffs?

L102: there was a wash out period? indicate and the length

L103-104: This is confusing; the forage was wet with water? this is a normal management in donkeys feeding?

L104-105: the rations were formulated to satisfy maintenance requirements? Include this information. Also please include if the animals were ad libitum or not

L108: You must include the quantity of incubated feeds, the ratio incubated gram: bag area is a key factor in techniques including bags.

L116: It is a common procedure to frozen the bags and then wash them, in order to eliminate better attached microorganisms.

L122: indicate procedures references

L127: the title of this subsection is confusing. It would be suitable “calculations and statistical analysis”

Line 147-148. The observed increasement was statistically significant? I don’t see the time factor in the statistical model. You also don´t mention contrast analysis in the statistical model to analyse linearity or quadratic changes with time. You can´t say that the changes are linear…

The cecal sampling did not last 13 h?, why did you show BC at 10.5 h?

L150: The rank is not correct. MF and HF are nor statistically different nor HF and LF

L153-158: the type of incubated feed was not in the statistical model. The authors may describe and discuss only statistically significant differences. You can´t discuss these ranks

Line 159-164. Please indicate the D1/D2 ratio in the Table and indicate if there are significant differences.

Discussion

Discussion should be improved because does not succeed in discussing scientifically the observed findings in a biologically integrated fashion, both within the study as well as relative to results of other scientists.

L180: you can´t say that the time affected BC, because you have not included time in the statistical model

L180-187 explain in which species were studied in these references.

L188: there were not differences between MF an LF. You can´t say this

L191: what are the implications of these results?

L194-205: This paragraph is too long; you are only explaining the methodology. In my opinion the discussion should be more centred in the observed results and its implications

L211-219: in the statistical analysis you didn’t include the effect of incubated feed, or at least you did not mention in the mat and met description of the statistical model. Therefore, you cannot discuss these numerical differences.

L220: rewrite

L220-233: in the statistical analysis you didn’t include the effect of incubated feed, or at least you did not mention in the mat and met description of the statistical model. Therefore, you cannot discuss these numerical differences.

L230-233: this is a conclusion of your study????

Author Response

This paper deals with the digestibility evaluation of 4 feedstuffs under three diet managements using the mobile bag technique in donkeys. The mobile bag technique has been used during decades for digestibility assays with different animal species and digestibility values for the studies feedstuffs has been analysed elsewhere. Therefore, the novelty of this paper is not very high apart from the animal species used.

Overall the paper is well written, but English should be revised. Only significant results should be address and not discuss numerical changes. The statistical model should be improved. Discussion must be improved.

Introduction

Point 1:  In the introduction there is a lack of background arguments to hypothesize that the forage to concentrate ratio could affect retention time or digestibility of tested feeds.

 You should include a cleared objective for your study. You commented in the intro that the problem is that the rations actually are formulated following NRC, which assumes some digestibility values and in donkeys would be different. Therefore, I don´t understand the objective of the study because digestibility references were obtained under standard feeding regimes, and there is no justification for the inclusion of different forage to concentrate ratios.

Response 1: Following your suggestions, the introduction section has revised again, and hopefully the objective is reasonably described.

Point 2:  Materials and methods

 The four feedstuffs included in the study were usually used in donkey’s rations? They are the most usually used feedstuffs?

Response 2: Yes, these four feedstuffs are the most commonly used in donkey production in Xinjiang.

Point 3L102: there was a wash out period? indicate and the length

Response 3: Yes, length of bag introduction and bag collection has been noted now.

 Point 4:  L103-104: This is confusing; the forage was wet with water? this is a normal management in donkeys feeding?

Response 4: The forage was chopped into approximately 1 cm lengths and then wetted by adding 30% water. Afterwards, the wetted forage was mixed with the concentrate to prepare total mixed rations. Yes, this is a normal management in donkeys feeding

 Point 5:  L104-105: the rations were formulated to satisfy maintenance requirements? Include this information. Also please include if the animals were ad libitum or not

Response 5: According to feed intake measurement prior to the start of the experiment, approximately 1.2 times amount of each experimental ration was given for ad libitum feeding. The description of feed allowance has been revised and hopefully it should be clear now.

Point 6:  L108: You must include the quantity of incubated feeds, the ratio incubated gram: bag area is a key factor in techniques including bags.

Response 6:  Regarding the forage of rice straw and alfalfa hay, 500 mg feed sample was weighed into mobile nylon bags. Regarding corn meal and soybean meal, 700 mg feed sample was weighed into mobile nylon bags.

 Point 7:  L116: It is a common procedure to frozen the bags and then wash them, in order to eliminate better attached microorganisms.

Response 7: Yes, we indeed stored at -20C for later laboratory washing procedure.

 Point 8:  L122: indicate procedures references

Response 8: Yes, the reference has been added now.

 Point 9:  L127: the title of this subsection is confusing. It would be suitable “calculations and statistical analysis”

Response 9: Yes, it has revised following your suggestion.

 Point 10:  Line 147-148. The observed increasement was statistically significant? I don’t see the time factor in the statistical model. You also don´t mention contrast analysis in the statistical model to analyse linearity or quadratic changes with time. You can´t say that the changes are linear…

Response 10: Yes, the description has been revised now

 Point 11:  The cecal sampling did not last 13 h?, why did you show BC at 10.5 h?

Response 11: The collection description has been revised now as : After 1.5 h of the bag introduction, the lips of fistulas were opened once an hour until 10.5 h and , and the bags arrived near the ileocecal junction were captured with the magnetic bar and recorded to determine pre-cecum digestibility (D1).

 Point 12:  L150: The rank is not correct. MF and HF are nor statistically different nor HF and LF

Response 12: Yes, the result description has been corrected now.

 Point 13:  L153-158: the type of incubated feed was not in the statistical model. The authors may describe and discuss only statistically significant differences. You can´t discuss these ranks

Response 13: The difference between feed type is obvious, so we did not include feed type in the statistical model. Following your comments, we have revised the paragraph by removing the rank description now.

 Point 14:  Line 159-164. Please indicate the D1/D2 ratio in the Table and indicate if there are significant differences.

Response 14: Yes, no significant difference between diet treatments occurred for the digestion contribution of pre-caecal tract in total digestive tract (D1/D2).

Discussion

Point 15:  Discussion should be improved because does not succeed in discussing scientifically the observed findings in a biologically integrated fashion, both within the study as well as relative to results of other scientists.

Response 15: The previous study in donkeys are quite limited, anyway, we now revised the discussion to the issues you mentioned, hopefully improvements should have been somewhat made now.

 Point 16:  L180: you can´t say that the time affected BC, because you have not included time in the statistical model

Response 16: Ok, that description has been deleted now

 Point 17:  L180-187 explain in which species were studied in these references.

Response 17: The species of animals been added now.

Point 18:  L188: there were not differences between MF an LF. You can´t say this

Response 18: Yes, the description has been revised now

 Point 19:  L191: what are the implications of these results?

Response 19: Yes, the implication has been added now

 Point 20:  L194-205: This paragraph is too long; you are only explaining the methodology. In my opinion the discussion should be more centred in the observed results and its implications

Response 20: Yes, the descripting explaining the methodology has been deleted now.

 Point 21:  L211-219: in the statistical analysis you didn’t include the effect of incubated feed, or at least you did not mention in the mat and met description of the statistical model. Therefore, you cannot discuss these numerical differences.

Response 21: Although the effect of incubated feed was included in the statistical analysis, the obvious numerical differences could somewhat explain that such difference varied mainly depending on the type of feed.

 Point 22:  L220: rewrite

Response 22: Yes, it has been revised now.

 Point 23:  L220-233: in the statistical analysis you didn’t include the effect of incubated feed, or at least you did not mention in the mat and met description of the statistical model. Therefore, you cannot discuss these numerical differences.

Response 23: As explained in the above, Although the effect of incubated feed was included in the statistical analysis, the obvious numerical differences could somewhat explain that such difference varied mainly depending on the type of feed

 Point 24:  L230-233: this is a conclusion of your study????

Response 24: Ok, the description has been deleted now.

Reviewer 3 Report

Li-Lin liu et al. Animals-775716: “Effect of dietary forage: concentrate ratio on pre-2 caecal and total digestive tract digestibility of diverse 3 feedstuffs in donkeys as measured by the mobile 4 nylon bag technique”.”.

This is an interesting study that evaluated the caecal and total tract digestibility of different feedstuffs in Donkeys. This is as interesting topic and can bring valuable information to improve the nutritive value of an important feedstuff for donkey and horse nutrition used in China. The manuscript is written and seems to have used appropriate methods and experimental design, I only have a few comments to improve the MS, they are listed below:

Line 18. "donkeys was measured"

Lines 22-23. This is conclusion is too vague. Could you be more specific?? E.g., What did you mean by Botanical origin??

Line 103. "the animals had free access"

Line 108. "A replicate 3 x 3 Latin square experimental"

Line 111. " the dietary treatments included:"

Line 116. "Dietary treatments"

Line 142. "Neutral Detergent Fibre"

Line 181. "three dietary treatments". Please, correct this accordingly along the entire MS.

Line 207. I am surprise that the HF diet had higher flow rate compared to other treatments. Usually, in ruminants, high fiber diets have higher retention time and lower passage rate compared to medium and low fiber diets...

Table 3. Describe the S.E.M in the footnote of the table.

Author Response

Line 18. "donkeys was measured"

Response: Yes, accepted and revised

Lines 22-23. This is conclusion is too vague. Could you be more specific?? E.g., What did you mean by Botanical origin??

Response: The nutrient composition of the feed, especially the fiber content was the main factor that affects the digestibility contribution of the prececum.

Line 103. "the animals had free access"

Response: Yes, accepted and revised

Line 108. "A replicate 3 x 3 Latin square experimental"

Response: Yes, accepted and revised

Line 111. " the dietary treatments included:"

Response: Yes, accepted and revised

Line 116. "Dietary treatments"

Response: Yes, accepted and revised

Line 142. "Neutral Detergent Fibre"

Response: Yes, accepted and revised

Line 181. "three dietary treatments". Please, correct this accordingly along the entire MS.

Response: Yes, accepted and revised

Line 207. I am surprise that the HF diet had higher flow rate compared to other treatments. Usually, in ruminants, high fiber diets have higher retention time and lower passage rate compared to medium and low fiber diets...

Response: This is probably caused by the difference in digestive physiology of equine animals and ruminants, which I will further verify in future research.

Table 3. Describe the S.E.M in the footnote of the table.

Response: Yes, accepted and added now

Round 2

Reviewer 1 Report

I have no more comments.

Author Response

Thank you for your valuable comments and support.

Reviewer 2 Report

The authors have addressed some of my comments. However, there are some comments that should be addressed before considering publication.

L17:19: revise the English, rewrite.

L20-21: As I mentioned in the previous revision the type of incubated feed was not in the statistical model, so you can´t infer this type of comparisons. Although the numerical changes could seem obvious only statistical significant differences must be discussed. Otherwise, if you think that it is of relevance, you can include this variable in the statistical model and discuss the obtained results.

L37: delete step-wise

L41: diets or dietary treatments

L 41-43: As I mentioned in the previous review, the type of incubated feed was not in the statistical model. The authors may describe and discuss only statistically significant differences. You can´t discuss these ranks

L50-53: Please rewrite this sentence, it is not very clear.

L52-54: What do you want to say with the “forage fibre content”?. The forage to concentrate ratio?, or the type of feed?? (you tested not only forages). This sentence is confusing. If you are referring to the type of feed, take into account that you have not include the type of incubated feed in the statistical model. If you wanted a conclusion at this regard please include it.

L51 and 79: Not foregut. It should be pre-caecum?

L61-63: please revise the English and rewrite.

L 81: this sentence needs a citation.

L101: and each animal was housed in a separate pen with free access to water

L103-104: revise English

L115: there was a wash out period? indicate and the length. The wash out period is a timeframe between periods when animals were fed a standard diet, in order not to confound treatment effects.

L117: delete and

L130-131: the remaining bags were collected in feces to ….

L134: I suppose that you have weighted the bags after washing them. Please include this information

L139: include AOAC method number for each determination.

L147: feed

L559-162: You could include the type of incubated feed in the statistical model

L166: As I commented in the previous review, the rank is not correct. MF and HF are nor statistically different nor HF and LF

173-179: Please include the type of feed in the statistical model in order to state this. Moreover, revise the English in L177-179.

L204-207: There were no differences between HF and LF in BC, neither between HF and MF. How can you state this??? Your BC results suggest that the recovery rate of bags will increase when donkeys were fed high-fibre ration???

L210: rumen not ruminants

213-216: Revise the English. i.e: observed in horses, observed in ponies…

L222: there is any reference in literature to the lower mean retention time with high forage diets in hindgut fermenters?. Please include

L242-244 and elsewhere in the discussion: I repeat, to say this you have to include type of incubated feed in the statistical model.

L257 and L260 and elsewhere in the document: not foregut, use pre-cecal

263-266: revise English and rewrite

268: there is no conclusion related to the objective of the study

Table 1: Alfalfa hay had a surprisingly low CP content

Author Response

The authors have addressed some of my comments. However, there are some comments that should be addressed before considering publication.

L17:19: revise the English, rewrite.

Response: Yes, it has been revised now

L20-21: As I mentioned in the previous revision the type of incubated feed was not in the statistical model, so you can´t infer this type of comparisons. Although the numerical changes could seem obvious only statistical significant differences must be discussed. Otherwise, if you think that it is of relevance, you can include this variable in the statistical model and discuss the obtained results.

Response: Yes, the incubated feed effect has been included in the statistical model, corresponding results have been added into Table 4 though the table in comparison with previous version gets somewhat complicated.

L37: delete step-wise

 Response: Yes, it has been deleted

L41: diets or dietary treatments

Response: Yes, it has been deleted

L 41-43: As I mentioned in the previous review, the type of incubated feed was not in the statistical model. The authors may describe and discuss only statistically significant differences. You can´t discuss these ranks

Response: Yes, the incubated feed effect has been included in the statistical model.

L50-53: Please rewrite this sentence, it is not very clear.

Response: Yes, it has been revised

L52-54: What do you want to say with the “forage fibre content”?. The forage to concentrate ratio?, or the type of feed?? (you tested not only forages). This sentence is confusing. If you are referring to the type of feed, take into account that you have not include the type of incubated feed in the statistical model. If you wanted a conclusion at this regard please include it.

Response: Yes, it has been revised.

L51 and 79: Not foregut. It should be pre-caecum?

Response: Yes, it has been revised

L61-63: please revise the English and rewrite.

Response: Yes, it has been revised

L 81: this sentence needs a citation.

Response: Yes, it has been revised

L101: and each animal was housed in a separate pen with free access to water

Response: Yes, it has been added now

L103-104: revise English

Response: Yes, it has been revised

L115: there was a wash out period? indicate and the length. The wash out period is a timeframe between periods when animals were fed a standard diet, in order not to confound treatment effects.

Response: Regarding the Latin square experimental design, three consecutive periods (3´25 days) were arranged as following table, for each period, it includes 18 days for diet adaption, 3 days for the introduction of mobile nylon bags and 4 days for the collection of all bags. There was no ‘washout period’ as you said. When one period was completed, the animals were then shifted to next diet that they never ate previously following the Latin square design.

Animal 1 and animal 2

Animal 3 and animal 4

Animal 5 and animal 6

Period1

HF

MF

LF

Period2

MF

LF

HF

Period3

LF

HF

MF

L117: delete and

Response: Yes, it has been deleted now.

L130-131: the remaining bags were collected in feces to ….

Response: Yes, it has been revised

L134: I suppose that you have weighted the bags after washing them. Please include this information

Response: Yes, you are right.

L139: include AOAC method number for each determination.

Response: Yes, it has been revised

L147: feed

L559-162: You could include the type of incubated feed in the statistical model

Response: Yes, it has been added now.

L166: As I commented in the previous review, the rank is not correct. MF and HF are nor statistically different nor HF and LF

Response: Yes, it has been deleted now. In the previous version, we want to mention, at the collection time point of 10.5, the rank is correct, anyway, following you comment, we have deleted it now.

173-179: Please include the type of feed in the statistical model in order to state this. Moreover, revise the English in L177-179.

Response: Yes, it has been added now.

L204-207: There were no differences between HF and LF in BC, neither between HF and MF. How can you state this??? Your BC results suggest that the recovery rate of bags will increase when donkeys were fed high-fibre ration???

L210: rumen not ruminants

Response: Yes, it has been revised

213-216: Revise the English. i.e: observed in horses, observed in ponies…

Response: Yes, it has been revised

L222: there is any reference in literature to the lower mean retention time with high forage diets in hindgut fermenters?. Please include

Response: Yes, it has been revised

L242-244 and elsewhere in the discussion: I repeat, to say this you have to include type of incubated feed in the statistical model.

Response: Yes, it has been added now.

L257 and L260 and elsewhere in the document: not foregut, use pre-cecal

Response: Yes, all have been revised

263-266: revise English and rewrite

Response: Yes, it has been added now.

268: there is no conclusion related to the objective of the study

Table 1: Alfalfa hay had a surprisingly low CP content

Response: Yes, this is our measured value. This is what we planted in the experimental field. Due to poor control of planting technology and harvesting and drying, the nutritional value is not high.